# Boosting Humoral Immunity from mRNA COVID-19 Vaccines in Kidney Transplant Recipients

**DOI:** 10.3390/vaccines10010056

**Published:** 2021-12-31

**Authors:** Leszek Tylicki, Alicja Dębska-Ślizień, Marta Muchlado, Zuzanna Ślizień, Justyna Gołębiewska, Małgorzata Dąbrowska, Bogdan Biedunkiewicz

**Affiliations:** 1Department of Nephrology, Transplantology and Internal Medicine, Faculty of Medicine, Medical University of Gdańsk, 80-210 Gdańsk, Poland; adeb@gumed.edu.pl (A.D.-Ś.); marta.muchlado@gumed.edu.pl (M.M.); zuzanna.slizien@gumed.edu.pl (Z.Ś.); jgolebiewska@gumed.edu.pl (J.G.); bogdan.biedunkiewicz@gumed.edu.pl (B.B.); 2Central Clinical Laboratory, University Clinical Centre, 80-952 Gdańsk, Poland; m.dabrowska@uck.gda.pl

**Keywords:** COVID-19, kidney transplant recipients, mRNA vaccines, seroconversion

## Abstract

**Introduction:** The immune response to the primary (two-dose) series of mRNA COVID-19 vaccines in kidney transplant recipients (KTRs) is very weak. We conducted a longitudinal observational study to compare the humoral response to a third, additional primary dose of mRNA vaccines between infection-naïve (IN-KTRs) and previously infected KTRs (PI-KTRs). Methods: We measured the levels of anti-spike (anti-s) IgG antibodies before and 14–21 days after the third dose and, in the secondary analysis, we compared the antibody response to BNT162b2 versus mRNA-1273. The reactogenicity assessment included solicited local and systemic reactions. **Results:** A total of 112 KTRs were enrolled, including 83 IN-KTR and 29 PI-KTR, among whom seroconversion in anti-s antibodies after the primary two-dose vaccination was achieved in 45.78% and 100% of cases, respectively. After three months, a waning antibodies titer by 67.4% (IN-KTR) and 7.5% (PI-KTR) was observed. After the third dose of the mRNA vaccine, 71.08% (59/83) of IN-KTR and 96.5% (28/29) of PI-KTR samples were seroconverted with a median anti-s titer of 468.0 (195.0–1620.0) BAU/mL and 1629.0 (1205–1815) BAU/mL, respectively. Of those IN-KTR in whom the primary vaccination failed, 46.67% (21/45) of patients achieved seroconversion after the third dose. No serious adverse events after the third dose were reported. In strata analyses, after the third dose, 66% (40/60) of patients vaccinated with BNT162b2 and 82.6% (19/23) of patients vaccinated with mRNA-1273 seroconverted with a median anti-s titer of 384.5 (144–837) BAU/mL and 1620 (671–2040) BAU/mL, respectively. **Conclusions**: The use of a third dose of mRNA vaccine may be of benefit for KTR, especially for those in whom the primary vaccination failed. Vaccines with a higher dose of mRNA and a longer interval between doses of the primary vaccination, such as mRNA-1273, seem to be the preparations of choice in immunocompromised individuals.

## 1. Introduction

Two 2-dose messenger RNA (mRNA)-based vaccines, mRNA-1273 and BNT162b2, were the first vaccines to receive Emergency Use Authorization (EUA) in the US and Europe. Both of these vaccines encode the prefusion-stabilized full-length spike (s) protein of severe acute respiratory syndrome coronavirus 2 (SARS-CoV-2), also known as COVID-19, but they differ in terms of mRNA content (100 μg for mRNA-1273 vs. 30 μg for BNT162b2), the interval between priming and boosting doses (4 weeks for mRNA-1273 vs. 3 weeks for BNT162b2), and the lipid composition of the nanoparticles used for packaging the mRNA content. As indicated by the registry studies, the effectiveness of both vaccines exceeds 90% and gradually decreases over time in the following months due to the waning of neutralizing antibodies [1].

The efficacy of SARS-CoV-2 vaccination among solid organ transplant recipients (SOTRs) is low [2]. In kidney transplant recipients (KTRs), we previously demonstrated a markedly impaired seroconversion rate of 51.4% for neutralizing antibodies after two doses of an mRNA vaccine. In addition, the magnitude of antibody-mediated response to vaccination in seroconverted individuals was much lower than that in immunocompetent controls [3]. Stumpf et al. not only showed a low antibody-mediated response following two doses of mRNA vaccines but also demonstrated a substantial impairment of the cell-mediated response [4]. In light of the very high mortality rate of COVID-19, up to 30% in some studies [5], the Polish Minister of Health issued a recommendation in September 2021 that immunocompromised patients should receive a “booster” vaccination but that it should not be sooner than 28 days after completion of the primary series. The present study reports on the antibody-mediated response in KTRs who have received a third dose of an mRNA SARS-CoV-2 vaccine.

## 2. Materials and Methods

### 2.1. Study Population

This longitudinal observational study was performed in a group of KTRs whose condition is managed at our institution and have received the primary vaccination series consisting of two doses of an mRNA vaccine, either BNT162b2 (Comirnaty, Pfizer/BionTech) or mRNA-1273 (Moderna), given according to the manufacturer’s recommendations as described previously [3]. The third dose of the vaccine (the additional primary dose) was administered immediately following the recommendation of the Polish Ministry of Health. The patients were therefore vaccinated three months after the second dose of the primary vaccination with a standard single injection of an mRNA vaccine. Subjects received the same vaccine as in the primary series. Patients with known SARS-CoV-2 infection in the past were also vaccinated according to the same rules. The study included those patients who presented for the third dose of the vaccine between 30 October 2021 and 9 November 2021.

### 2.2. Study Design

The main aim of the study was to compare the seroconversion rates and titers of IgG antibodies to the SARS-CoV-2 S protein after the third dose of an mRNA vaccine between infection-naïve KTRs (IN-KTRs) and previously infected KTRs (PI-KTRs). Serum samples for anti-S antibodies were obtained 3 months after the second dose (just before the booster) and 14–21 days after the third dose of the vaccine (Figure 1). The nucleocapsid (N)-specific IgG antibody serostatus was checked to exclude or confirm a prior asymptomatic SARS-CoV-2 infection. The secondary endpoints included: solicited common and expected adverse reactions shortly (i.e., within 7 days) after the third dose of the vaccine (reactogenicity), and unsolicited adverse events and serious adverse events, i.e., those reported by the participants without prompting from the medical staff or observed by their physicians through a period of 1 month after the third dose. In strata, secondary analyses were performed for the comparison of antibody response following vaccination with BNT162b2 and mRNA-1273. The study was approved by the Ethics Committee of the Medical University of Gdańsk (resolution NKBBN/167/2021) and conducted in accordance with the Declaration of Helsinki. The study is part of the COVID-19 in Nephrology (COViNEPH) project, which focuses on the nephrological aspects of COVID-19; in particular, its epidemiology, prevention, clinical course, and treatment.

### 2.3. Procedures and Analytical Methods

A quantitative determination of specific IgG antibodies to the trimeric S-protein located on the virus’s outer surface was performed using a commercial chemiluminescent immunoassay kit (The LIAISON^®^ SARS-CoV-2 TrimericS IgG test, DiaSorin, Italy, catalog number: P/N311510, webpage) as described previously [3]. The assay presented a sensitivity of 98.7%, a specificity of 99.5%, and agreement with neutralization in microneutralization tests: PPA: 100%, NPA: 96.9%. Neutralizing antibodies (NAbs) are defined as an antibody that defends a cell from a pathogen or infectious particle by biologically neutralizing any effect that it has. The presence of NAbs is commonly considered a sign of protection against a pathogen. Samples were interpreted as positive (seroconversion) or negative (no seroconversion) with a cutoff index value of >33.8 BAU/mL, according to the manufacturer. A positive result indicates the presence of IgG S-antibodies to SARS-CoV-2 and indicates exposure to SARS-CoV-2 or a humoral response to vaccination. A negative result may indicate the absence or a very low level of IgG antibodies to the pathogen. The same analytical method was used to assess the presence of seroconversion and antibody titers.

N-specific IgG antibodies were assessed with a commercial quantitative chemiluminescent immunoassay kit (SARS-CoV-2 IgG, Abbott Laboratories, USA, catalog number: 6R86-20, webpage). N-protein is present in the viral core and plays a vital role in viral transcription. Natural exposure induces a dominant antibody response against the N protein, but since N protein is not in the vaccine, there is no vaccine-induced response against it. Determining only the titer of anti-S antibodies did not allow us to differentiate whether the humoral response was due to disease or vaccination. Samples were interpreted as positive (seroconversion) with a signal/cutoff (S/C) index value of 1.4.

Reactogenicity data were obtained through interviews performed by health staff according to a standardized questionnaire, as described previously [6]. The grading scales were derived from the FDA Center for Biologics Evaluation and Research (CBER) guidelines on toxicity grading scales for healthy adult volunteers enrolled in preventive vaccine clinical trials. The assessments included solicited local (pain, redness, swelling) and systemic reactions (fever, fatigue, headache, chills, vomiting, diarrhea, new or worsened muscle pain, and new or worsened joint pain). Serious adverse events were defined as any untoward medical occurrence that resulted in death, was life-threatening, required inpatient hospitalization or the prolongation of existing hospitalization, or resulted in persistent disability/incapacity. The medical records of patients from the Outpatient Department were analyzed in order to confirm prior SARS-CoV-2 infection.

### 2.4. Statistical Analyses

Due to the skewed distribution of most continuous variables, these are expressed as medians (interquartile ranges [IQR]). Categorical variables are presented as counts (percentages). Continuous variables were first tested for normal distribution using the Shapiro–Wilk test, and then compared using the *t*-test, if normally distributed, and by the Mann–Whitney or Wilcoxon test where appropriate, if non-normally distributed. The chi-square test was used for categorical variables. Data were analyzed with Statistica (version 12.0, Stat Soft, Inc., Dell Software, Tulsa, OK, USA). Those *p*-values of <0.05 were considered statistically significant.

## 3. Results

### 3.1. Patient Characteristics

A total of 112 KTRs were enrolled in the study. Their characteristics are provided in Table 1. The cohort was stratified, based on the evidence of a previous SARS-CoV-2 infection, into 83 IN-KTRs (with no history of COVID-19 and negative test results for N-specific antibodies) and 29 PI-KTRs (with a history of COVID-19 and/or positive test results for N-specific antibodies). IN-KTR and PI-KTR did not differ in terms of age, sex, graft function, graft vintage, and type of immunosuppression. A total of 45.78% (38 of 83 patients) of IN-KTRs and 100% (29 of 29 patients) of PI-KTRs were positive for anti-S antibodies immediately after the primary vaccination, with median titers of 365.3 (117.3–915.2) BAU/mL and 1219.4 (647.4–1978.6) BAU/mL, respectively.

### 3.2. Anti-S IgG Antibodies before and after the Third mRNA Vaccine Dose

Three months after the second dose of the mRNA vaccine (just before the third dose), 37.25% (31/83) IN-KTRs and 96.5% (28/29) PI-KTRs were seroconverted, with median anti-S titers of 119.0 (81.1–265.0) BAU/mL and 1128.0 (380.0–4280) BAU/mL, respectively (Figure 2).

After the third dose of the mRNA vaccine, 71.08% (59/83) of IN-KTRs and 96.5% (28/29) of PI-KTRs were seroconverted, with median anti-S titers of 468.0 (195.0–1620.0) and 1629.0 (1205–1815) BAU/mL, respectively (Figure 2). Of the IN-KTRs whose primary vaccination failed, 46.67% (21/45) achieved seroconversion with a median antibody titer of 155 (55.9–361) BAU/mL.

In strata analyses performed in IN-KTRs, seroconversion after the second dose of the mRNA vaccines was observed in 38.33% of patients (23/60) who had received BNT-162b2, and in 65.22% (15/23) of those vaccinated with mRNA-1273 (*p* = 0.028). After the third dose, seroconversion was observed in 66.67% (40/60) and 82.61% (19/23) individuals, respectively (*p* = 0.15). In seroconverted IN-KTRs, median anti-S antibody titers after the second dose of BNT162b2 and mRNA-1273 were 213.2 (81.2–548.6) and 717.6 (260–936) BAU/mL (*p* = 0.07), respectively. After the third dose of BNT162b2 and mRNA-1273, median antibody titers were 384.5 (144–837) and 1620 (671–2040) BAU/mL (*p* = 0.002), respectively (Figure 3).

### 3.3. Reactogenicity to the Third mRNA Vaccine Dose

Of all KTRs, 63.39% (71/112) reported at least one local site reaction within 7 days of the third dose of the mRNA vaccine. They reported only mild to moderate injection-site reactions. No grade-3 or -4 local reactions were reported. Pain at the injection site was the most frequent local reaction among the vaccines. The median duration of local reactions was 2.25 (1–2) days. At least one of the solicited systemic reactions occurred in 23.21% (26/112) of KTRs. The most frequent solicited systemic reactions were fatigue, followed by muscle pains, headaches and joint pains (Figure 4). The majority of patients reported only mild to moderate systemic reactions. One patient (0.009%) had severe systemic symptoms in the form of a high fever. No grade-4 systemic reactions were reported. The median duration of systemic symptoms was 1 (1–4.5) days. No serious adverse events following the vaccination were reported. In strata analyses, there were no significant differences in the frequency of local and systemic reactions between the IN-KTRs (62.54% (52/83) and 19.28% (16/83), respectively) and PI-KTRs (65.52% (19/29) and 34.48% (10/29), respectively). There were no significant differences in the frequency of local and systemic reactions between those vaccinated with BNT162b2 (61.18% (52/85) and 20% (17/85), respectively) and mRNA-1273 (70.37% (19/27) and 33.33% (9/27), respectively).

## 4. Discussion

Our study is one of the first few studies that show the immunogenic potential of an additional primary mRNA vaccine dose in immunocompromised patients after transplantation [7,8,9]. As in other studies, fewer than 50% of the patients in our cohort showed any antibody-mediated response following the primary vaccination, consisting of two doses. We have also shown a very rapid waning of the antibody response in the infection-naïve responders, specifically, an antibody titer decrease of over 67% within 3 months of completing the vaccination procedure. The third dose produced a large increase in the number of antibodies, achieving a titer almost 30% higher than that after the second dose of the vaccine. Most importantly, seroconversion was achieved in almost half of the patients in whom the primary vaccination failed. The results of our study are consistent with several previous observations made in heterogeneous groups of SOTRs. In the first randomized controlled trial, in 120 SOTRs, a third mRNA-1273 vaccine dose resulted in substantially higher immunogenicity than the placebo. The seroconversion rate was 55% in the booster group vs. 18% in the placebo group [10]. Kamar et al. showed that the administration of a third dose of BNT162b2 to 101 SOTRs significantly improved the antibody-mediated response to the vaccination. Among the 59 patients who had been seronegative after the standard two-dose vaccination with BNT162b2, 26 (44%) were seropositive at 4 weeks after the third dose [11]. In another study, among 232 SOTRs who were seronegative before the third dose, 105 (45.25%) turned seropositive after the third dose of BNT162b2 [12]. Observations in the population of KTRs are scarce. Among 10 KTRs who failed seroconversion after the primary series of mRNA-1273 vaccinations, the antibody- and cell-mediated responses after the third dose were elicited in 60% and 90% of patients, respectively [9]. Another study showed that the third dose of mRNA-1273 induced a serologic response in 49% of 97 KTRs who had not responded after 2 doses [7].

Similar to the primary vaccination, a stronger antibody response after a booster dose occurred in subjects who received mRNA-1273 [3]. This is evidenced by the higher seroconversion rate, reaching 83%, and more than four times higher antibody titer (Figure 3). Similar observations have been made in the general population [13,14]. Moreover, although both mRNA vaccines are highly effective, the latest reports from the first head-to-head study indicate a slightly better 24-week effectiveness of mRNA-1273 than BNT162b2, regardless of the predominant strain—Alpha, earlier and then Delta, later [1]. There are two potential explanations for this fact. Firstly, mRNA-1273 contains a higher dose of mRNA. Each dose of BNT162b2 delivers only 30 µg of mRNA, while each dose of the mRNA-1273 vaccine contains 100 µg [15]. Secondly, the interval between the first and the second dose differs between the two vaccines: the injections are 21 days apart for BNT162b2 and 28 days apart for mRNA-1273. Some studies have shown that longer time intervals between the primary and booster doses are associated with higher effectiveness [16,17].

In line with the previous observation in various populations, patients with a prior SARS-CoV-2 infection responded better to vaccination [18,19,20]. This applies not only to the development of antibody-mediated response but also to its magnitude. The results seem to prove that exposure to the virus (its whole particle) is a stronger stimulus than vaccination in terms of immune memory development and antibody formation upon re-exposure to the antigen. This is the most complete form of immunization, which seems to provide the currently optimal protection against severe COVID-19, and this is called hybrid immunity [21]. Hybrid immunity, at least in part, is mediated by memory B cells. The majority of antibodies formed after infection or vaccination come from short-lived cells called plasmablasts, and antibody levels fall when these cells inevitably die off. Once they are gone, memory B cells become the main source of the antibodies and these are induced by infection or vaccination [22]. Importantly, 3 months after the primary vaccination, the antibody titer among prior-infected KTR decreased slightly and, in contrast to infection-naïve KTR, remained still at a high level. The dose of the booster significantly increased this titer further. Although it is known that the higher the titer, the lower the risk of developing COVID-19, considering the fact that KTR are a group at high risk of death, it seems risky to skip a booster dose in prior-infected KTR.

Although we observed a significant antibody response in the majority of KTRs, nearly one-third of the patients still failed to produce neutralizing antibodies. The effectiveness of the fourth dose of the mRNA vaccine in this group of patients requires verification. The few exploratory studies do not provide a clear answer to this question, indicating the legitimacy or illegitimacy of such an action [23,24]. The primary heterologous vaccine schedules or a heterologous booster after wild-type primary vaccination also seem worthy of further research [25,26]. How the observed difference in seroconversion and antibody titer translates to protection against COVID-19, the protection against variants of concern, and the difference in the duration of protection needs further investigation as well. The recent, large observational study, conducted using nationwide mass vaccination data from Israel, demonstrated that a third dose of the BNT162b2 mRNA vaccine is effective in preventing severe COVID-19-related outcomes in the general population [27]. When analyzing our research, one must bear in mind its limitations, including the observational design of the study and the fact that T-cell tests were not performed.

In conclusion, the use of a third dose of mRNA vaccines is of benefit in kidney transplant recipients, especially those in whom the primary vaccination failed. Vaccines with a higher dose of mRNA and a longer interval between the primary doses, such as mRNA-1273, seem to be the products of choice in immunocompromised individuals.

## Figures and Tables

**Figure 1 vaccines-10-00056-f001:**
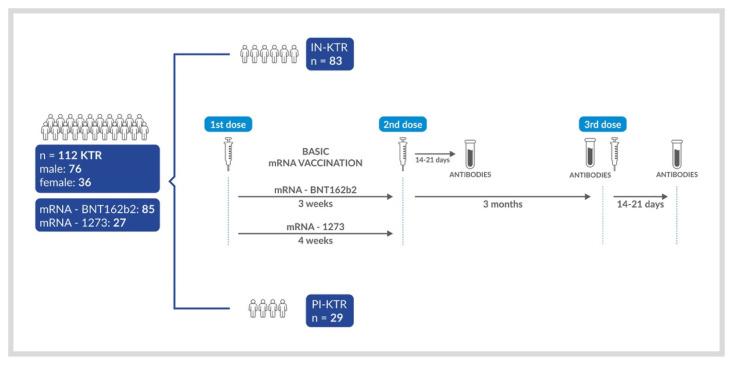
Graphical scheme of the study.

**Figure 2 vaccines-10-00056-f002:**
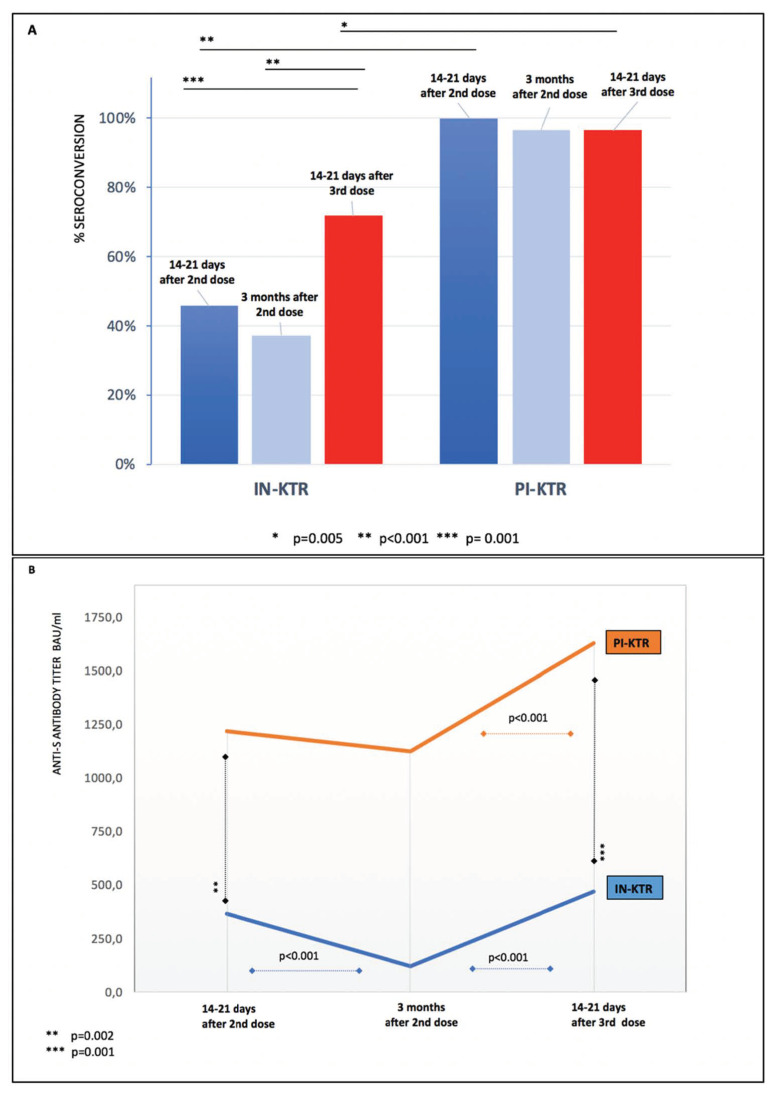
Anti-S IgG antibody response after mRNA SARS-CoV-2 vaccination in infection-naïve (IN) and previously infected (PI) kidney transplant recipients (KTRs); (**A**): seroconversion rate; (**B**): antibody titer (seroconversion rate and antibody titer were analyzed with a LIAISON^®^ SARS-CoV-2 TrimericS IgG test).

**Figure 3 vaccines-10-00056-f003:**
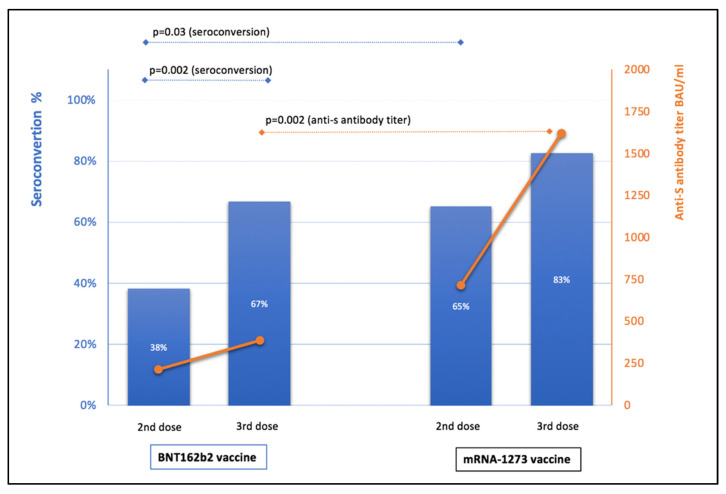
Seroconversion for anti-S IgG antibodies (blue bars) and median anti-s IgG titer (orange lines) in IN-KTR after the second and third mRNA vaccine doses.

**Figure 4 vaccines-10-00056-f004:**
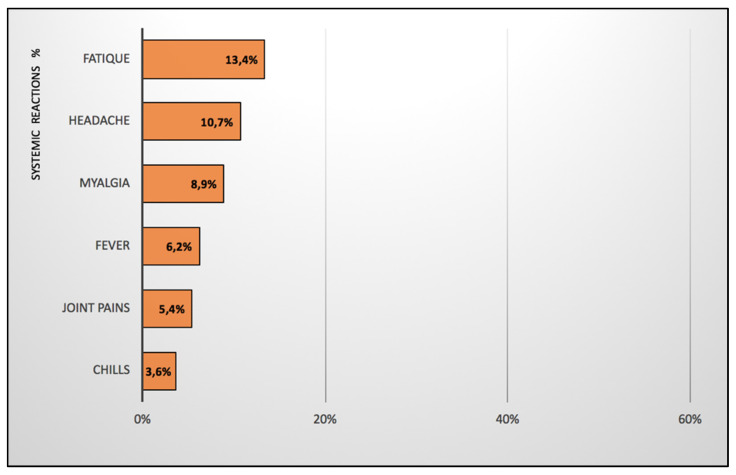
Reactogenicity (systemic reactions) to the third mRNA vaccine dose.

**Table 1 vaccines-10-00056-t001:** Patient characteristics.

	KTRs, Total n = 112	IN-KTR n = 83	PI-KTR n = 29	*p*-Value **
Age (years)	54.5 (44.5–63)	55 (42–63)	54 (48–61)	0.95
Male sex Female sex	76 (67.86) 36 (32.14)	54 (65.06) 29 (34.94)	22 (75.86) 7 (24.14)	0.28 0.28
CCI	4 (2–6)	4 (2–6)	4 (2–6)	0.74
Serum creatinine (mg/dl)	1.34 (1.13–1.68)	1.35 (1.13–1.6)	1.33 (1.19–1.8)	0.46
BMI (kg/m^2)^	25.9 (22.59–28.96)	25.98 (22.69–28.96)	25.43 (22.55–28.37)	0.71
Transplant duration (years)	8 (3.5–15)	8 (3.5–15)	8 (3.0–15)	0.81
Deceased donor	102 (91.07)	75 (90.36)	27 (93.1)	0.66
Immunosuppression protocol Protocol without steroids Protocol without MMF/MPS	10 (8.93) 26 (23.21)	9 (10.84) 18 (21.89)	1 (3.45) 8 (27.59)	0.23 0.52
mRNA-1273 vaccination	27 (24.11)	23 (27.71)	4 (13.79)	0.13
mRNA BNT162b2 vaccination	85 (75.89)	60 (72.29)	25 (86.21)	0.13
Seroconversion after second dose ^a^	67 (59.82)	38 (45.78)	29 (100)	< 0.001
Anti-S antibody titer after second dose ^a,^*	663.0 (213.2–1560)	365.3 (117.3–915.2)	1219.4 (647.4–1978.6)	< 0.001
Anti-S antibody titer 3 months after second dose *	265.0 (110.0–1040.0)	119.0 (81.1–265.0)	1128.0 (380.0–4280)	< 0.001

Abbreviations: CCI, Charlson comorbidity index; BMI, body mass index; MMF/MPS, mycophenolate mofetil/sodium. Data are expressed as medians (interquartile ranges [IQR]) for continuous variables and counts (percentages) for categorical variables. The seroconversion and anti-S antibody titer was analyzed by LIAISON^®^ SARS-CoV-2 TrimericS IgG test. ^a^ 14–21 days after the second dose. * Data for seroconverted individuals. ** Difference between IN-KTRs and PI-KTRs.

## Data Availability

Detailed data are available on request from the corresponding author.

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
