# Peer review of "Boosting Humoral Immunity from mRNA COVID-19 Vaccines in Kidney Transplant Recipients"

_vaccines, 2021, doi:10.3390/vaccines10010056_

Round 1

Reviewer 1 Report

 Reviewer’s Comments – Manuscript entitled “Boosting humoral immunity from
mRNA COVID-19 vaccines in kidney transplant recipients” by Tylicki et al. (Vaccines)

In the manuscript submitted by Tylicki and co-workers, the Authors have measured the spike-antibody responses in kidney transplant recipients (KTRs) after immunization with 2 doses of vaccine (BionTech/Pfizer or Moderna) and after receiving the third vaccine dose both in infection-naïve (IN-KTR) and in previously infected (PI-KTR) subjects. A total of 112 KTRs were included in the study. The results indicate that after two dose of vaccine the seroconversion rate was of approximately 50% among IN-KTR and 100% among PI-KTR. In addition, titres were significantly higher among PI-KTR and in general percentage of seroconversion was higher after Moderna vaccine immunization.  After the third vaccination, although few KTR subjects that were negative after two doses seroconverted, however nearly one third of KTRs did not produce neutralizing antibodies to vaccination after the third dose.  As concerned to reactogenicity to the third dose of vaccine no significant differences were reported among patients and the majority of subjects reported only mild-to-moderate systemic reactions.

The study addresses an important and challenging issue in the field of Sars-COV2 infection with possible beneficial impact on vaccination programs in KTRs and in general of immunocompromised subjects. The results fit the scope of the journal and may be of interest to the readership of Vaccines.

However, in my opinion, the Authors may improve the presentation of their results and the figures and should clarify few points.

Specific comments

Materials and methods

Section “Study population”: I think that a graphical scheme summarizing the study population (i.e n for vaccine type, time of vaccination, time of the third dose, bleeding time points, n of male and n of females…) would help the reader to better clarify the study scheme. In addition, it should be clarified if they were undergoing immune suppression therapy.

line 86: please specify what s/c stand for.

Table 1: why females’ characteristics were not included? Or were they all males? If this is the case, could you explain why females were not included?

Please correct “Anit-S antibody titers…” to “Anti-S …”

Please specify in table and figure 1 when antibodies were tested after second dose. You report results of antibody titers (table) and seroconvertion (figure1)  afer second dose and after 3 months after the second dose. What is the difference?

Figure 2: please delete polish Obszar kreslenia above bar. Please indicate in the legend what is the line (I suppose is the median of the results).

Line 196: patients (not patietns)

The subjects which did not seroconverted were however analysed by B-cell elispot for the presence of memory B cells? It would be interesting to know if anyhow they had any type of priming of immune system in order to understand the reasons of their seronegative status and/or also to implement the vaccination program in KTRs.

Author Response

REVIEWER 1:

1. Section “Study population”: I think that a graphical scheme summarizing the study
population (i.e n for vaccine type, time of vaccination, time of the third dose, bleeding
time points, n of male and n of females...) would help the reader to better clarify the
study scheme. In addition, it should be clarified if they were undergoing immune
suppression therapy.
The Figure 1 with graphical scheme was added as requested (line 96). This caused a change
in the numbering of subsequent Figures

2. Line 86: please specify what s/c stand for.
It was done as requested (line 123).

3. Table 1: why females’ characteristics were not included? Or were they all males? If this
is the case, could you explain why females were not included?
It was done as requested (line 160).

4. Please correct “Anit-S antibody titers...” to “Anti-S ...”
It was done as requested.

5. Please specify in table and figure 1 when antibodies were tested after second dose.
You report results of antibody titers (table) and seroconvertion (figure1) afer second
dose and after 3 months after the second dose. What is the difference?
It was specified as requested (Table 1 and Figure 1) (line 96 and 160)

6. Figure 2: please delete polish Obszar kreslenia above bar. Please indicate in the legend
what is the line (I suppose is the median of the results).
It was done as requested in present Figure 2.
7. Line 196: patients (not patietns)
It was corrected.
8. The subjects which did not seroconverted were however analyzed by B-cell elispot for
the presence of memory B cells? It would be interesting to know if anyhow they had
any type of priming of immune system in order to understand the reasons of their
seronegative status and/or also to implement the vaccination program in KTRs.

We agree with the reviewer that such an analysis may have a very high clinical and scientific
value. Unfortunately, B cell analysis was not part of this study design. We plan to conduct
such research in the future.

Reviewer 2 Report

This manuscript, written by Dr. Leszek Tylicki, original research, with the title of “Boosting humoral immunity from mRNA COVID-19 vaccines in kidney transplant recipients” analyzes the seroconversion and antibody titers after vaccination in a group of patients with immunosuppression due to kidney transplantation. The research showed that after the third vaccine doses, in the infection-naïve group the levels increase. Therefore, that group benefits with the third doses. It seems that Moderna vaccine is more effective.

This manuscript is well written and describes important information that can benefit the patients with kidney transplantation. Before publishing these results, the authors may address the following comments/suggestions to improve the manuscript. 

Comments:

1) Line 53. Regarding “BNT162b2 (BionTech/Pfizer Comirnaty)”. I think that the proper name is “Comirnaty and Pfizer-BioNTech”.

2) Could you please comment if the efficacy of Pfizer–BioNTech COVID-19 vaccine is similar to the Moderna vaccine? A paragraph could be added in the introduction.

3) Could you please add information about the design of these two vaccines? In what aspect, they are similar, and what makes them different? For example, which subunit of the protein spike are they targeting, etc. This information could be added in the introduction.

4) Lines 99-100. Regarding the use of medians. Could you please explain why it was decided to use the medians in Table 1?

(Note: Both the mean and median can be used to describe where the “center” of a dataset is located. It’s best to use the mean when the distribution of the data values is symmetrical and there are no clear outliers. It’s best to use the median when the distribution of data values is skewed or when there are clear outliers).

4) Regarding the LIAISON® SARS-CoV-2 TrimericS IgG assay.

4.1) There are a series of clarifications that could be added in the text, so the reader should not need to refer to the previous publication, as follows:

“Neutralizing antibodies (NAbs) are defined as an antibody that defends a cell from a pathogen or infectious particle by neutralizing any effect it has biologically. 

The presence of NAbs is commonly considered a sign of protection against a pathogen

< 33.8 (Negative): A negative result may indicate the absence or a low level of IgG antibodies to the pathogen. The test could score negative in infected patients during the incubation period and in the early stages of infection.

≥ 33.8 (Positive): A positive result indicates the presence of IgG antibodies to SARS-CoV-2 

and indicates exposure to SARS-CoV-2.”

4.2) Line 83. Could you please define the word “seroconversion”? (This may help medical readers not specialist in this field but with interest in recent advances in coronavirus).

4.3) Line 80-84. Could you please explain why the Liaison test is used for determining the seroconversion and not the nucleocapsid-based test?

4.4) According to the text, the Liaison test is used both to confirm the percentage of seroconversion and the concentration of antibodies (anti-s igg antibody titer). Is this correct. In the text of the tables and figures, this information could be highlighted.

4.5) Could you please add the catalog number of this kit and the webpage link of this product? For instance, https://www.diasorin.com/sites/default/files/allegati_prodotti/liaisonr_sars-cov-2_trimerics_igg_assay_m0870004408_a_lr_0.pdf

5) Regarding lines 67-68, “Nucleocapsid (N)-specific IgG antibody serostatus was checked to exclude or confirm a prior asymptomatic SARS-CoV-2 infection”.

5.1). Lines 84-87. Could you please add the catalog number and the webpage link of this Abbot test?

5.2) Could you please specify if this test is quantitative or qualitative for detecting of IgG against the SARS-CoV-2 nucleoprotein (nucleocapsid antigen)?

5.3) Could you please provide more scientific background? For example, a brief description of the structure of the coronavirus, the spike, the nucleoprotein. 

5.4) In the lines 109-112 it is stated: “The cohort was stratified based on the evidence of a previous SARS-CoV-2 infection into IN-KTRs (with no history of COVID-19 and negative test results for N-specific antibodies) and PI-KTRs (with a history of COVID-19 and/or positive test results for N-specific antibodies).” Question: Could you please explain why the detection of antibodies against the nucleoprotein relates to the previous infection? Would not the Liaison test could be used for the same purpose? How was the history of COVID-19 assessed?

6) Do you have information about the results of both test at time 0 (before the first-second vaccination)?

7) Lines 131-134. Looking at the data looks like that the seroconversion is higher in the Moderna vaccine. Was the p value significant? Within each type of vaccine, there is an increase of seroconversion, was this increment statistically significant (two-related samples test of Wilcoxon)?

8) Lines 134-138 and Figure 2. Do this data of text and figure 2 refer to the IN-KTR group? Please confirm and specify in the text.

9) Figure 1A. Looking at the data, in PI-KTR the administration of a 3rd dose doesn’t make any effect. Therefore, these patients should not be vaccinated (only the IN-KTR group)?

10) Figure 1B. There is a difference between PI-KTR and INKTR. Have you performed statistics within each group using for example non-parametric test of two-related samples test of Wilcoxon? In the IN-KTR group the increase in titer doesn’t look significant. In the PI-KTR yes. 

11) Could you please show Figures 1A and 1B stratified for the vaccine types (in 4 figures)?

12) Regarding lines 223-224, “Vaccines with a higher dose of mRNA and a longer interval between the primary doses seem to be the products of choice”. I think this part refers to Figure 2. Maybe the data could be expressed as a in a rate of increase or something similar, with a p value?

13) Figure 2. “obszar kreslenia” (drawing area) should be erased from the figure.

14) Could you please show if the level of immunosuppression is similar between all patients?

Author Response

REVIEWER 2:
1. Line 53. Regarding “BNT162b2 (BionTech/Pfizer Comirnaty)”. I think that the proper
name is “Comirnaty and Pfizer-BioNTech”.
It was corrected as requested (line 69)
2. Could you please comment if the efficacy of PfizerBioNTech COVID-19 vaccine is
similar to the Moderna vaccine? A paragraph could be added in the introduction.
and
3. Could you please add information about the design of these two vaccines? In what
aspect, they are similar, and what makes them different? For example, which subunit of
the protein spike are they targeting, etc. This information could be added in the
introduction.
Ad 2 and 3: It was done as follows:
(Introduction line 41) “Two 2-dose the messenger RNA (mRNA) - based vaccines, mRNA-
1273 and BNT162b2, were the first ones which received Emergency Use Authorization (EUA)
in US and Europe. Both of these vaccines encode the prefusion stabilized full-length spike
protein of severe acute respiratory syndrome coronavirus 2 (SARS-CoV-2), but they differ in the
mRNA content (100 μg for mRNA-1273 vs. 30 μg for BNT162b2), the interval between priming
and boosting doses (4 weeks for mRNA-1273 vs. 3 weeks for BNT162b2), and the lipid
composition of the nanoparticles used for packaging the mRNA content. As indicated by the
registry studies, the effectiveness of both vaccines exceeds 90% and gradually decreases over
time in the following months due to the waning of neutralizing antibodies”

and

(Discussion- line 240). Moreover, although both mRNA vaccines are highly effective, the latest
reports from the first head-to-head study indicate a slightly better 24-week effectiveness of
mRNA-1273 than BNT162b2, regardless of the predominant strain -Alpha earlier and then Delta
later.

4. Lines 99-100. Regarding the use of medians. Could you please explain why it was
decided to use the medians in Table 1?
It was described as recommended in Statistics section (line 136).
5. Regarding the LIAISON® SARS-CoV-2 TrimericS IgG assay. There are a series of
clarifications that could be added in the text, so the reader should not need to refer to
the previous publication, as follows:
“Neutralizing antibodies (NAbs) are defined as an antibody that defends a cell
from a pathogen or infectious particle by neutralizing any effect it has
biologically.
The presence of NAbs is commonly considered a sign of protection against a
pathogen

< 33.8 (Negative): A negative result may indicate the absence or a low level of
IgG antibodies to the pathogen. The test could score negative in infected
patients during the incubation period and in the early stages of infection.
≥ 33.8 (Positive): A positive result indicates the presence of IgG antibodies to
SARS-CoV-2 and indicates exposure to SARS-CoV-2.”
The detailed description was included to the methods section as recommended (line 104).
6. Line 83. Could you please define the word “seroconversion”? (This may help medical
readers not specialist in this field but with interest in recent advances in coronavirus).
It was done as requested in method section (line 109).
7. Line 80-84. Could you please explain why the Liaison test is used for determining the
seroconversion and not the nucleocapsid-based test?
It was clarified in the method section (line 121)
8. According to the text, the Liaison test is used both to confirm the percentage of
seroconversion and the concentration of antibodies (anti-s igg antibody titer). Is this
correct. In the text of the tables and figures, this information could be highlighted.
The same analytical method (the Liason test) was used to assess the presence of
seroconversion and antibody titers. It was mentioned in the methods section, in table and
figure (line 114, Table 1 and Figure 2).
Could you please add the catalog number of this kit and the webpage link of
this product?
It was done as requested (line 104)
9. Regarding lines 67-68, “Nucleocapsid (N)-specific IgG antibody serostatus was
checked to exclude or confirm a prior asymptomatic SARS-CoV-2 infection”.
Lines 84-87. Could you please add the catalog number and the webpage link of
this Abbot test?
Could you please specify if this test is quantitative or qualitative for detecting of
IgG against the SARS-CoV-2 nucleoprotein (nucleocapsid antigen)?
It was done as requested in method section (line 117)
Could you please provide more scientific background? For example, a brief
description of the structure of the coronavirus, the spike, the nucleoprotein.
It was done as recommended in the method section (line 118)
In the lines 109-112 it is stated: “The cohort was stratified based on the
evidence of a previous SARS-CoV-2 infection into IN-KTRs (with no history of
COVID-19 and negative test results for N-specific antibodies) and PI-KTRs (with
a history of COVID-19 and/or positive test results for N-specific antibodies).”
Question: Could you please explain why the detection of antibodies against the
nucleoprotein relates to the previous infection? Would not the Liaison test
could be used for the same purpose? How was the history of COVID-19
assessed?

It was clarified in methods section as follows: Natural exposure induces a dominant
antibody response against the N protein, but since N protein is not in the vaccine,
there is no vaccine induced response against it. Determining only the titer of anti-S
antibodies did not allow to differentiate whether the humoral response was due to
disease or vaccination (line 118)
Medical records of patients from the Outpatient Department were analyzed in order to
confirm prior SARS-COV-2 infection” (line 133)

10. Do you have information about the results of both test at time 0 (before the first-second
vaccination)?
Antibody analyzes between the first and second primary vaccine doses were not performed

11. Lines 131-134. Looking at the data looks like that the seroconversion is higher in the
Moderna vaccine. Was the p value significant? Within each type of vaccine, there is an
increase of seroconversion, was this increment statistically significant (two-related
samples test of Wilcoxon)?
The p-values (when significant) were added to the Figure 3 (line 187)
12. Lines 134-138 and Figure 2. Do this data of text and figure 2 refer to the IN-KTR group?
Please confirm and specify in the text.
Its true. It was specified in the text (line 178) and actual Figure 3 legend (line 187).

13. Figure 1A. Looking at the data, in PI-KTR the administration of a 3rd dose doesn’t make
any effect. Therefore, these patients should not be vaccinated (only the IN-KTR group)?
I think we know too little to draw such conclusions. First of all, the titer of antibodies protecting
patients against COVID-19 / severe course is not known. It is known, however, that antibody
levels are waning over time in all patients. The third dose does not change the seroconversion
rate in Pi-KTR (because the seroconversion cut-off titer is set at a fairly low level) but
significantly increases the s-antibody titer (1128vs.1629 BAU). Taking into account that the
higher the titer, the lower the risk of developing COVID-19 disease / severe course, and the
fact that KTR are a group at high risk of death, it seems risky to skip a booster dose in PI-
KTR. Some comment in this regard has been added to the discussion (line 262-267)

14. Figure 1B. There is a difference between PI-KTR and INKTR. Have you performed
statistics within each group using for example non-parametric test of two-related
samples test of Wilcoxon? In the IN-KTR group the increase in titer doesn’t look
significant. In the PI-KTR yes.
The p-values were added to the actual Figure 2B (line 172)

15. Could you please show Figures 1A and 1B stratified for the vaccine types (in 4
figures)?

The analysis and figure proposed by the Reviewer would be interesting and desirable in a
study with a larger group of patients. Unfortunately, in our case, after two-fold stratification
according to the type of vaccine and then on the prior-infection incidence, subgroups with too
small numbers of cases were formed (eg. n=4 for PI-KTR vaccinated with mRNA-1273);
therefore no such analysis was performed.

16. Regarding lines 223-224, “Vaccines with a higher dose of mRNA and a longer interval
between the primary doses seem to be the products of choice”. I think this part refers
to Figure 2. Maybe the data could be expressed as a in a rate of increase or something
similar, with a p value?
The p values of the differences between the humoral response after the two vaccines have
been added to the actual Figure 3 (line 187) and the results section (line 178). The differences
between vaccines demonstrated in the study were emphasized more strongly in the
discussion with reference to Figure 2 (line 238)
17. Figure 2. “obszar kreslenia” (drawing area) should be erased from the figure.
It was done in actual Figure 3.

18. Could you please show if the level of immunosuppression is similar between all
patients?
Levels of immunosuppressants in blood were not performed for the purpose of the study.
Patients from both subgroups did not differ in the type of immunosuppression and the time after
transplantation (Table 1). This is highlighted in the results section (line 150). Therefore, it can
be assumed that the level of immunosuppression was similar in both groups.